# The Interactions of Food Security, Health, and Loneliness among Rural Older Adults before and after the Onset of COVID-19

**DOI:** 10.3390/nu14235076

**Published:** 2022-11-29

**Authors:** Mecca Howe-Burris, Stacey Giroux, Kurt Waldman, Julia DeBruicker Valliant, Angela Babb, Kamila Czebotar, Daniel Fobi, Phil Stafford, Daniel C. Knudsen

**Affiliations:** 1Department of Anthropology, Indiana University, 701 E Kirkwood Ave., Bloomington, IN 47405, USA; 2Ostrom Workshop, Indiana University, 513 N Park Ave., Bloomington, IN 47408, USA; 3Food Institute, Indiana University, 405 N Park Ave., Bloomington, IN 47408, USA; 4Department of Geography, Indiana University, 701 E Kirkwood Ave., Bloomington, IN 47405, USA; 5Department of Anthropology, Indiana University, 5598 East Ward Lane, Bloomington, IN 47408, USA

**Keywords:** rural, older adults, path analysis

## Abstract

Older adults and those living in rural areas face unique challenges to health and food security which were exacerbated during the COVID-19 pandemic. We examine the interrelationships among food security, physical health, and loneliness among rural older adults before the onset of and during the COVID-19 pandemic. Using data from a cross-sectional household survey of older adults in rural Indiana, administered May 2020 through July 2020, our results show a higher prevalence of food insecurity, poorer physical health, and increased loneliness after the onset of the pandemic. Path analyses confirmed the interrelationship between food security, health, and the absence of loneliness before and after the onset of COVID-19. Loneliness emerged as a major pathway through which the pandemic impacted quality of life, particularly affecting older women and physical health. Policy initiatives should consider the unique experiences and challenges associated with rural life among older adults and create food security initiatives that incorporate socialization while also considering the challenges associated with poor physical health in older age. Furthermore, our analysis shows that those who are vulnerable to food security, poor health, and loneliness in the absence of a global pandemic remain vulnerable during a pandemic.

## 1. Introduction

The COVID-19 pandemic has intensified disparities in well-being among older adults, particularly those who were food insecure before the onset of the pandemic [1]. Food insecurity, the inability to consistently access safe and nutritious foods for health and wellbeing [2], often coexists with poor health, isolation, and loneliness, particularly for those living in rural areas. While the pandemic increased rates of food insecurity, poor health, and loneliness throughout the country, rural communities were hit hard [1,3]. According to Feeding America, “rural communities have been particularly vulnerable during the pandemic because they frequently contain older populations, higher rates of chronic disease, and limited healthcare infrastructure” [3]. Shutdowns and stay-at-home orders, while important for mitigating the spread of the virus, played a role in increasing food insecurity, isolation, and feelings of loneliness [1,4,5]. Congregate meal sites, including many senior centers, were shut down due to the pandemic [6]. This took away one of the few, if not the only, opportunities for older adults in rural spaces to socialize and share a reduced-cost meal with others [6,7]. Likewise, the closing of food pantries, food shortages, limited store and pantry hours, and fear of contracting the virus further impacted food security [4,8].

Because food insecurity contributes to stress, inflammation, and malnutrition, the increase in food insecurity among rural older adults during the COVID-19 pandemic also increased vulnerability to chronic disease and poor mental health [9]. Furthermore, chronic diseases like diabetes and cardiovascular disease increase one’s risk for COVID-19 complications. In addition, pandemic-related loneliness has been associated with poorer physical and mental health, worsened food security, and an increase in poor dietary behaviors such as emotional eating and unhealthy snacking [1,5].

The COVID-19 pandemic further challenged the quality of life of older adults, particularly in rural communities. In this paper, we use survey data and path analysis to explore the interrelationships of food security, physical health, and the absence of loneliness among rural older adults before and after the onset of the pandemic. Then, we examine the impact of the COVID-19 pandemic on quality of life by comparing changes to food security, physical health, and the absence of loneliness before COVID-19 and after the onset of COVID-19 using a cross-sectional, retrospective household survey of older adults in four rural counties in Indiana.

### 1.1. Rural Challenges for Older Americans

More than 72% of communities in the United States (U.S.) are considered rural [10]. Rural counties are defined by the U.S. Office of Management and Budget and include towns with fewer than 25,000 people as well as small cities that are not part of a larger metropolitan area and have fewer than 50,000 people [10]. The median percentage of older adults in completely rural counties is more than 20%, while in urban centers, older adults account for approximately 13.8% of the population [11]. Rural populations are disproportionately impacted by food, health, and psychosocial disparities. Rural counties have higher rates of poverty (15% in rural places vs. 12% in urban) as well as food insecurity (13% vs. 11%) when compared to urban areas [3,12]. Moreover, rural counties account for 87% of U.S. counties with the highest rates of food insecurity [3]. Among adults 60+ years, non-metro residents have higher rates of food insecurity (7.3%) compared to metro older adults (6.7%) [13].

Spatial distance and poorer access to transportation negatively impact rural food security. In rural places, low-income households and households using the Supplemental Nutrition Assistance Program (SNAP)—the federal food assistance program that provides monthly income support to low-income households for the purchase of food—tend to live farther from the nearest food store than higher-income households not utilizing SNAP [14]. Longer distances to food stores are associated with lower food access and more food insecurity in rural areas, especially for people without personal transportation [6,15,16]. In rural Indiana in particular, grocery stores are not adequately located to serve households in poverty, especially those without vehicle access [17]. Furthermore, research shows rural areas have fewer food assistance sites with more limited hours and the nearest charity food assistance location is more than one mile away [18,19].

### 1.2. Connections between Food Security, Health, and Loneliness among Older Adults

Food insecurity is both indirectly and directly associated with distress and poor physical and mental health [20,21,22,23,24,25,26,27]. Indirectly, food insecurity is associated with lower household income, and lower household income is associated with poorer health outcomes [14,25,26]. Directly, physical impairments from chronic illness, disabilities, and/or aging make it difficult for residents to acquire, prepare, and eat certain foods, leading to a reliance on lower-quality meals and resulting in poorer nutrition and more food insecurity [9,26,27,28,29]. In addition, prescription medications can be a financial burden that leaves little money for food each month [6,25,29,30]. Older adults who are food insecure are more likely to have diabetes, hypertension, pulmonary disease, asthma, cardiovascular disease, heart attack, heart failure, depression, and obesity [20,26,27]. Food insecurity also associates with more physically unhealthy days, poorer health-related quality of life, disability, and the coexistence of multiple chronic conditions known as co-morbidities [25,28,31]. As a result, counties with the highest rates of food insecurity, which tend to be rural counties, also have high rates of chronic disease and sickness [31].

Health is an important aspect of perceived quality of life, which deteriorates with age and the prevalence of chronic disease. According to the Centers for Disease Control (CDC), 80% of older adults have a chronic disease such as type-2 diabetes, cardiovascular disease, or hypertension, and 33% experience challenges in daily activities such as preparing meals [32]. More than 63% of older adults have two or more chronic conditions [33], and rural older adults make up 35% of this statistic. Co-morbidities have been found to further increase one’s risk for food insecurity, exacerbating the cycle of poor nutrition and poor health [25].

Loneliness and social interactions are also connected to perceived quality of life among older adults [34,35,36]. More social interactions correlate with better self-reports [34,35], while more loneliness and less social support associate with poorer physical and psychological health [36]. A lack of social interaction and/or loneliness, particularly related to eating meals, has been associated with a lack of desire to cook or eat full meals [37,38]. Instead, isolated and/or lonely older adults rely on an unbalanced diet of convenient processed foods and/or skip meals (e.g., frozen foods, snacks in place of meals) [1,6,37].Older adults who live alone, are divorced, never married, or widowed, feel lonely, or perceive themselves to have low social support are more likely to be food insecure [1,37,39]. For rural residents, the risk of loneliness and isolation is much higher as homes are farther apart and farther from the town center [39]. In addition, rural areas are IinI a rise in out-migration from rural to urban places [10,40], leaving rural older adults without family nearby [6]. In addition, rural spaces lack public transportation [6,15,18,41]. For individuals who cannot drive due to advanced age, physical impairments, and/or financial barriers, rural spaces can become even more isolating and access to food even more difficult [38,40]. Loneliness is also directly connected to health–adults who are lonely self-report poorer health and have higher rates of chronic illnesses [36,42].

## 2. Materials and Methods

### 2.1. Defining Quality of Life

The focus on quality of life in this analysis stems from previously published qualitative research findings [6]. The analytical framework comes from focus groups and in-depth interviews with older adults from our study area in southern Indiana (see Valliant et al., 2021 for a description of the qualitative findings). Focus groups and interviews showed that food security, health, and the absence of loneliness were the main drivers of perceived quality of life for older rural adults in our sample. Food insecurity, on the other hand, was a cause of stress and worry not only about having enough food or getting the right kinds of food but also about the stigma and shame that can accompany food insecurity and/or limited income. Poor physical health was distressing and anxiety-inducing, and participants explained how economic burden, feeling unwell, and mobility challenges related to chronic health conditions [6]. Lastly, lack of social interaction induced feelings of loneliness, isolation, and sadness resulting in a lower perceived quality of life. These findings informed the conceptualization of quality of life used in the present study.

### 2.2. Hypotheses

The literature shows food insecurity, poor health, and loneliness are often experienced in tandem among older adults and can lead to coping mechanisms that inadvertently maintain or exacerbate these conditions. As a result, we expected that food security, physical health, and the absence of loneliness would positively associate with one another—where being food secure would predict more physically healthy days and less loneliness; more physically healthy days would predict greater food security and less loneliness; and the absence of loneliness would predict greater food security and more physically healthy days.

We also explore how food security, health, and loneliness varied by demographics, income, and isolation. We hypothesized that age, marital status, income, travel times to stores, and eating with others would be associated with food security as well as physical health. We anticipated that loneliness (or its absence) would be associated with age, gender, marital status, income, travel times to stores, and eating with others.

Furthermore, we analyzed how these interactions and overall quality of life may have changed as a result of the COVID-19 pandemic to gauge the impact of the pandemic on rural older adults. We expected to find higher rates of food insecurity, fewer physically healthy days, and more loneliness after the onset of COVID-19. In addition, we expected stronger interrelations among the outcome variables based on the literature that shows food insecurity is related to chronic disease risk and COVID-19 risk and complications. Furthermore, COVID-19 can further impair physical health leading to more severe food insecurity and isolation [4,9].

### 2.3. Research Site

The study area consisted of four rural counties in south-central Indiana. Table 1 describes the basic demographics of the counties. The counties, on average, had higher poverty rates (11–17%) than the national rate of 11%, as well as a large older adult population (20–21% compared to the national 17%; [43]. There are approximately 20,904 older adults (60+ years) across the four counties. More than 13% have incomes below the poverty threshold [43], and 54%, on average, have incomes below the SNAP threshold of 130% poverty [31]. Approximately 8400 (40.2%) older adults are within the lowest two quintiles of household income. The average food insecurity rate across the counties was 14.8% in 2019, and 14.3% in 2020 [31]. In comparison, the national rates of food insecurity were 11.7% in 2019 and 10.9% in 2020 [13]. Previous research has also found the region to be underserved by emergency food assistance programs [19].

### 2.4. Data Collection

A paper survey was mailed to households with older adults (defined as aged 60 years and above) across the four study counties using an address-based sampling frame purchased from Marketing Systems Group. For the sample, we stratified by county, households with incomes ≤ 185% of the poverty threshold, and persons 60 years of age or older. We then targeted census block groups within those counties with the highest percentages of poverty. The survey was sent to a sample of 5000 older adult households. Surveys were preceded by a pre-survey informational letter and followed by a reminder postcard one week after the survey was mailed. Surveys included a $5 grocery gift card incentive for completion. The surveys collected data on sociodemographics, household composition, dietary restrictions of household members, provisioning strategies (e.g., food sources, modes of transportation, food assistance participation), food security tabulated by the USDA Six-item Household Food Security Survey Module (HFSSM) (2012) [44], health indexed by the Healthy Days Core Model Questionnaire (HRQOL-4) [45], and loneliness indexed by the UCLA Three-Item Loneliness Scale [46]. The Six-item HFSSM captures food (in)security at the household level by asking questions pertaining to food access, quality, and quantity over the last 12 months [44]. The HRQOL-4 includes four questions pertaining self-reported health quality and the number of days within the last 30 days that one’s physical and mental health were not good [45]. The UCLA Three-Item Loneliness scale has three questions that evaluate one’s perceived relational and social connectedness and isolation [46].

Surveys were mailed in May 2020 and asked participants to respond to questions about their behavior and experiences before the COVID-19 pandemic and since the onset of the pandemic. A total of 1482 completed surveys were returned for a participation rate of 29.6%.

### 2.5. Data Analysis

All statistical analyses were conducted using STATA 16. Paired t-tests were used to compare mean household size, travel times to stores, food security score, number of physically healthy days in the last thirty days, and loneliness scores before and after the onset of the COVID-19 pandemic. Chi-square was used to compare income levels, SNAP participation, eating with others, and food insecurity rates before and after the onset of COVID-19.

We used path analysis to explore the complex relationships among food security, health, and loneliness, as well as the effects of sociodemographic variables on these outcomes. Path analysis is a valuable tool when trying to understand a complex social phenomenon because it allows analysis of the relationships between multiple dependent variables as well as the relationships between independent and dependent variables in a single analytical model [47]. In addition, path analysis permits the inclusion of different types of regression analyses simultaneously [47]. In our path analyses, we used non-parametric linear regression to test the relationships between continuous variables due to the non-normal distribution of our outcome variables. We used logistic regression when the dependent variables were categorical.

To determine which independent variables to include in the path analysis, we conducted univariate (also known as unadjusted or simple) regression analyses with each of the dependent variables (full results upon request). Variables that were significantly (α = < 0.05) associated with food security, health, and loneliness in the simple regression models were included in the path analysis. Table 2 provides a list of variables and their definitions included in the path analysis models. For food security, physically healthy days, and loneliness, inverse scores were used to ensure increasing scores would indicate better outcomes (better food security, more healthy days, and less loneliness) to reflect quality of life (a positive notion) more easily.

Travel times between homes and food stores were quantified using ARCGIS and geocoding home addresses and grocery store locations in the home county as well as the adjacent counties. Secondly, Network Analyst was used to calculate the travel time in minutes between each respondent’s home address and each store. The sum of the travel times from each respondent’s home address to each grocery store provided a total one-way travel time from each respondent’s home address to all grocery stores in the home county as well as the adjacent counties.

We conducted three path analyses: (1) one with variables pertaining to before the onset of COVID-19 (variables denoted with “BC” for “before the onset of COVID-19”), (2) a second with variables relating to after the onset of COVID-19 (variables denoted with “AC” for “after the onset of COVID-19”), and (3) a third model to understand how food security, health, and loneliness before COVID-19 impacted food security, health, and loneliness after the onset of COVID-19.

## 3. Results

General sample characteristics are described in Table 3, and Table 4 compares before and after the onset of COVID variables. All the quality-of-life variables were significantly impacted by the onset of the pandemic. T-tests showed the mean food security score significantly declined after the onset of COVID (Table 4) meaning the number of people who were food insecure significantly increased after the onset of the pandemic. The number of physically healthy days significantly declined after the onset of COVID, and older adults reported feeling significantly lonelier during the pandemic as compared to before COVID-19. Income among this population increased for those at the lower end of the scale and increased for those at the upper end. Few people participated in SNAP and far fewer people ate with others after the onset of the pandemic.

### 3.1. Before COVID Path Analysis

Before the onset of COVID-19, older age and increasing income were positively associated with food security while increasing travel time was negatively correlated with food security (see Figure 1 and Table A1 of the Appendix A). However, the coefficient and confidence intervals for travel times were close to zero, indicating a weak association. More healthy days and less loneliness also predicted greater odds of food security.

The number of physically healthy days was negatively correlated with increasing age. Being food secure, less lonely, and having higher income positively correlated with more physically healthy days.

Increasing travel times were associated with more loneliness, while living with a partner or spouse, eating with others, being food secure, and having more healthy days were all associated with less loneliness.

### 3.2. Changes during COVID-19

This analysis examined how the same predictor variables related to outcomes after the onset of the pandemic (Table A2 in the Appendix A). Figure 2 highlights only the differences in relationships observed after the onset of COVID when compared to the before-COVID model (to make it easier to see the changes; a graph of the full path analysis results including all relationships after the onset of COVID can be found in Figure A1 of the Appendix A). All independent variables that were significantly associated with food security before COVID-19 (age, income, and travel times) remained significantly related in the after-COVID model. Physically healthy days also remained a significant predictor of food security. However, the absence of loneliness is no longer significantly associated with food security after the onset of the pandemic.

All independent variables that significantly correlated with physically healthy days before COVID-19 (age and income) remained significant predictors of healthy days after the onset of COVID. Loneliness (or the absence of loneliness) became more strongly related to physically healthy days after the onset of the pandemic, where more loneliness was associated with fewer physically healthy days during COVID-19.

Marital status, eating with others, food security, and health remained significantly associated with the absence of loneliness after the onset of COVID. While age and gender were not significantly related to loneliness before COVID, they became significant predictors of the absence of loneliness in the after-COVID models. Older age and identifying as male predicted less loneliness after the onset of the pandemic. On the other hand, travel times were not significantly related to loneliness after the onset of COVID but were significantly related before COVID.

### 3.3. Controlling for Before-COVID Food Security, Health, and Loneliness

When including before-COVID food security, health, and loneliness, no sociodemographic variables remained significant predictors of food security or physically healthy days after the onset of COVID-19 (see Figure 3; for full path analysis results including all associations see Figure A2 and Table A3 of the Appendix A). Only before-COVID food security and after-COVID health and loneliness were significantly associated with food security after the onset of the pandemic. Similarly, only before-COVID health and after-COVID food security and loneliness predicted physically healthy days after COVID. On the other hand, while after-COVID loneliness (and the absence of loneliness) was significantly impacted by before-COVID loneliness, age, eating with others, and gender remained significant predictors of loneliness after the pandemic.

## 4. Discussion

Among this sample, overall quality of life including food security, physical healthy days, and loneliness worsened with the onset of the COVID-19 pandemic. Compared to before the pandemic, two months into the 2020 shutdown period, the 1400+ rural older adults who responded to the survey were statistically likely to report feeling less food secure, less physically healthy, and lonelier. These findings are complementary to previous studies. The COVID Impact Survey found that perceived physical health decreased and feelings of loneliness increased during the COVID-19 pandemic and more older adults (65+ years) more often worried about getting enough food at the start of the pandemic. Using data from the Health and Retirement Study, Ankunda et al., (2021) found 20% of older adults (51+ years) were food insecure in November of 2020 but food insecurity rates decreased with age, suggesting that the oldest Americans were more resilient or otherwise protected [48]. In contrast, our sample of rural older adults became more food insecure during the pandemic, despite decreasing food insecurity rates found regionally and nationally [13]. This may be explained, in part, by the fact that only 30 participants participated in SNAP, meaning most of our sample did not experience the advantages and padding of increased SNAP benefits (Indiana SNAP benefits were increased to the maximum allotment per household size in March 2020 [48]).

Other factors beyond income were just as, or more, important in determining the likelihood of food insecurity among our sample of rural older adults, indicating that food insecurity is part of a more complex issue that extends beyond socioeconomic status. Both before and after the onset of the COVID-19 pandemic, food security, health, and loneliness were strongly interrelated. After the onset of the pandemic, food security was less strongly influenced by income and age, and the relationship of loneliness to other factors changed. With loneliness more widespread during the pandemic, our data show that loneliness was no longer associated with food insecurity and women were more susceptible to loneliness. Loneliness became more strongly related to physically healthy days. Older age was protective against loneliness and travel times to stores was not significantly related to loneliness after the onset of the pandemic. These findings may be explained by the low variation in perceived loneliness among the sample, as more people, including those of younger ages, reported feeling lonely after the pandemic began.

When controlling for before COVID-19 quality of life outcome variables, our data show that pre-COVID-19 food security, physical health, and loneliness were the most influential factors for quality of life during the COVID-19 pandemic. In this model, sociodemographic variables became insignificant. This posits that vulnerabilities before the pandemic were the most influential factors associated with food insecurity, poor health, and loneliness during the pandemic.

### 4.1. Food Insecurity and Health

We found strong evidence that food security is both influenced by and an influencer of physical health and the absence of loneliness. On the one hand, the cost of healthcare and medications can leave less money available for food, and physical impairments as the result of chronic disease can make it more difficult to obtain and prepare food [6,9,25,29,30,38]. This is supported by studies that show food insecurity is higher among those with physical disabilities [28]. In addition, individuals who do not feel well may have reduced appetites and/or experience mental health burdens that further limit the desire to eat [49,50,51]. A national analysis using the 2005–2016 National Health and Nutrition Examination Survey found food insecurity to be significantly associated with depression in U.S. adults [50]. During the pandemic, anxiety, illness, and poor emotional and mental health among older adults led to poorer appetites and poor nutritional behaviors, and, thus, could have reduced food security [51,52,53].

On the other hand, poor-quality diets related to food insecurity can contribute to chronic diseases and/or worsen the severity of illness [20,26,53,54]. Other studies have found that individuals who are food secure report more physically healthy days while those who are food insecure report more unhealthy days [21,23,24]. In addition, a recent analysis shows that older adults who are food insecure are more likely to skip or delay taking/refilling medications due to costs [54]. This can further inhibit physical health and increase poor physical health symptoms among older adults with chronic diseases like type 2 diabetes. Overall, food security is cyclically connected to better health-related quality of life, while food insecurity is cyclically connected to poorer health as evident by our analysis among rural older adults.

### 4.2. Food Insecurity and Loneliness

Before the onset of COVID-19, food security was significantly associated with loneliness where individuals who reported less loneliness were more food secure. This is consistent with Burris et al. (2019), which found loneliness and social isolation were significantly associated with higher odds of food insecurity among adults 65 years and older in Florida. Hanna and Collins (2015) note that older adults who live alone consume poorer-quality diets, especially less produce and fish, when compared to those who live with others [38]. This contributes to the higher rates of food insecurity commonly found among individuals who lack companionship. Marriage, in particular, is associated with eating with others and can be protective against loneliness and food insecurity [15,37,40].

During the pandemic, the relationship between food security and loneliness became insignificant among our sample when not controlling for pre-COVID food security and loneliness. This is likely due to the fact that the majority of respondents felt significantly lonelier and more reported eating their meals alone during the pandemic, including those who were food secure. It appears that other factors such as unique experiences associated with living through a fearful global pandemic (stay at home orders, fears of contracting the virus) and gender became more influential in whether an individual felt lonely or not. Increasing loneliness and isolation among older adults during COVID-19 has been documented in national and local studies [1,5]. This does not come as a surprise as older populations were among the most vulnerable to COVID-19 and, thus, were generally compliant with stay-at-home recommendations.

### 4.3. Loneliness, Health, and Gender

Loneliness increased as a result of the pandemic and became more strongly correlated with fewer physically healthy days. Loneliness often co-exists with, or exacerbates, mental and emotional health challenges like depression and cognitive decline [55]. However, loneliness can also be the result of physical chronic illness, disability, and cognitive decline as these conditions may limit one’s mobility and opportunities for socialization [42,55]. Studies show that individuals who perceive themselves to be lonely also report poorer health and higher rates of disability and chronic disease compared to those who are less lonely [5,42].

In addition, loneliness is often associated with poor nutrition which can subsequently lead to poor health outcomes or exacerbate chronic illnesses already experienced [1,37,39]. Schorr et al., 2020 found loneliness significantly related to malnutrition during the COVID-19 pandemic, and malnutrition is often the result of food insecurity [56]. Similarly, a Dutch study found older adults reported poorer nutritional behaviors including difficulty obtaining groceries, skipping meals or eating less than normal, and eating too little and/or losing weight during the COVID-19 pandemic in relation to staying home and more isolation [57]. Older adults who were in quarantine were even more likely to report worsened nutritional behaviors as well as significantly reduced physical activity [57].

Identifying as female became significantly related to more loneliness after the onset of the pandemic. This trend has been noted in nationally representative studies as well [5]. Previous literature has documented females as being more social and having wider social networks when compared to their male counterparts [35]. One study found that during the lockdown of COVID-19, females were more likely to suffer from mental health consequences than males [58]. Thus, the COVID-19 pandemic could have had less of an impact on older males who were more accustomed to social isolation and aloneness, while disproportionately impacting females who became separated from their social support networks. While females in this sample felt lonelier than males before the onset of the pandemic, the difference in loneliness scores was higher after the onset of COVID-19, indicating that females felt even more lonely than males during the pandemic. 

### 4.4. Limitations

Several known psychological factors likely impact the results. For example, some studies show older adults may underestimate or understate their level of food insecurity due to different perspectives of what it means to be food insecure [59,60], but also many food security surveys, including the USDA module, do not account for factors such as healthcare costs, chronic health conditions, or functional or cognitive limitations. Thus, the questionnaires themselves may underestimate food insecurity among older adults [34,59,60]. Furthermore, it is documented that individuals who identify as male may underreport poor mental, emotional, and/or physical health compared to females [11,61].

Our analysis is based on a single survey administered in May-July 2020, which requested that respondents answer questions concerning their current status and describe their pre-pandemic status in a typical month. Schmier and Halpren (2004) note that recall of health status can be unreliable, and, more generally, that recall bias typically occurs in cross-sectional, longitudinal studies [62]. We contend, though, that the short period between pre-COVID and during-COVID time frames in our study period (February to May 2020) would tend to minimize this bias.

Other limitations include that the survey was implemented fairly soon after the onset of the pandemic in the U.S. and therefore may underestimate the longer-term impacts on food security, health, and loneliness. Lastly, while the response rate of 29.6% is on the high end for postal surveys, the findings may not be representative of the whole population.

## 5. Conclusions

Rural communities are among the most vulnerable to negative impacts on quality of life, as demographic, spatial, and income inequalities are more prevalent [3,6,14,16,18,38]. Rural communities face unique challenges such as an increasingly aging population and subsequently high rates of chronic disease, higher rates of poverty, less access to transportation, physical isolation, and further distances to stores and food assistance [3,6,14,16,18]. Therefore, initiatives that may work in urban areas will not necessarily be successful if they do not consider the unique challenges, needs, and assets of rural communities.

In this study, we found that quality of life for rural older adults in Southern Indiana is strongly influenced by the intertwined relationships among food (in)security, physical health, and loneliness (or the absence of loneliness). Moreover, our data show that the issues of food insecurity, poor health, and loneliness are not experienced in isolation. We found that loneliness is a central pathway through which food security and health operate. The pandemic increased loneliness, especially among women in this sample, and created a stronger connection between loneliness and physically unhealthy days but a weaker connection between loneliness and food security. Thus, our data lend support for the fact that food insecurity is not just a nutritional and socioeconomic problem but is part of a multidimensional and complex experience. To improve quality of life and solve the issue of food insecurity among rural older adults, policies must also act to prevent and mitigate chronic disease and increase opportunities for social interaction that reduce feelings of loneliness.

## Figures and Tables

**Figure 1 nutrients-14-05076-f001:**
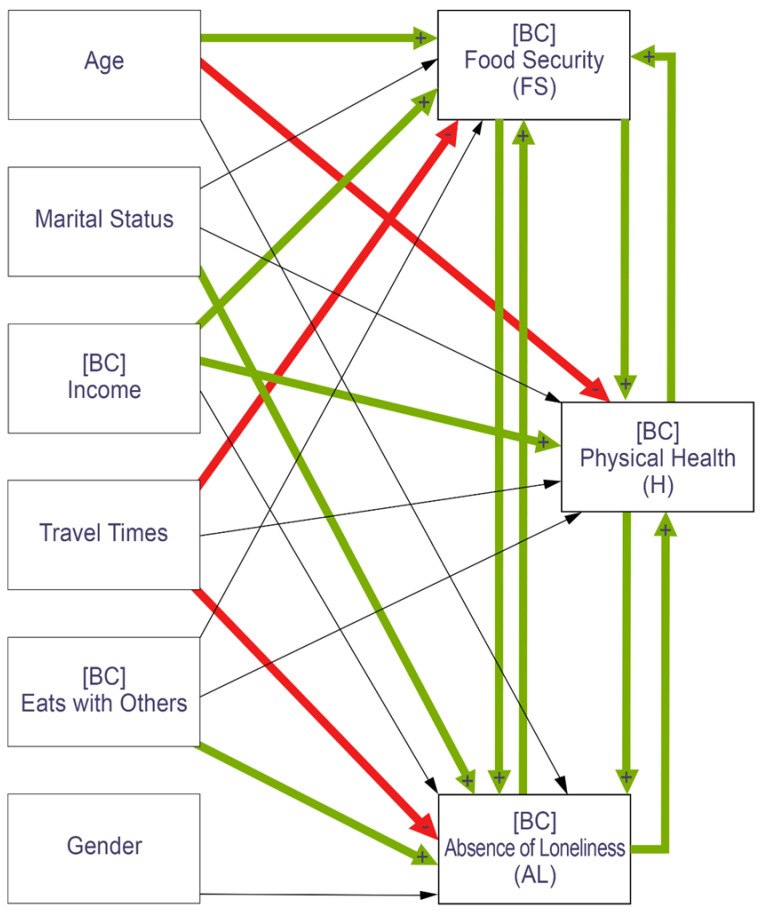
Path analysis for before the pandemic. Thick lines represent significance at α = 0.05. Thin lines represent insignificance. Green represents positive relationships and red negative relationships.

**Figure 2 nutrients-14-05076-f002:**
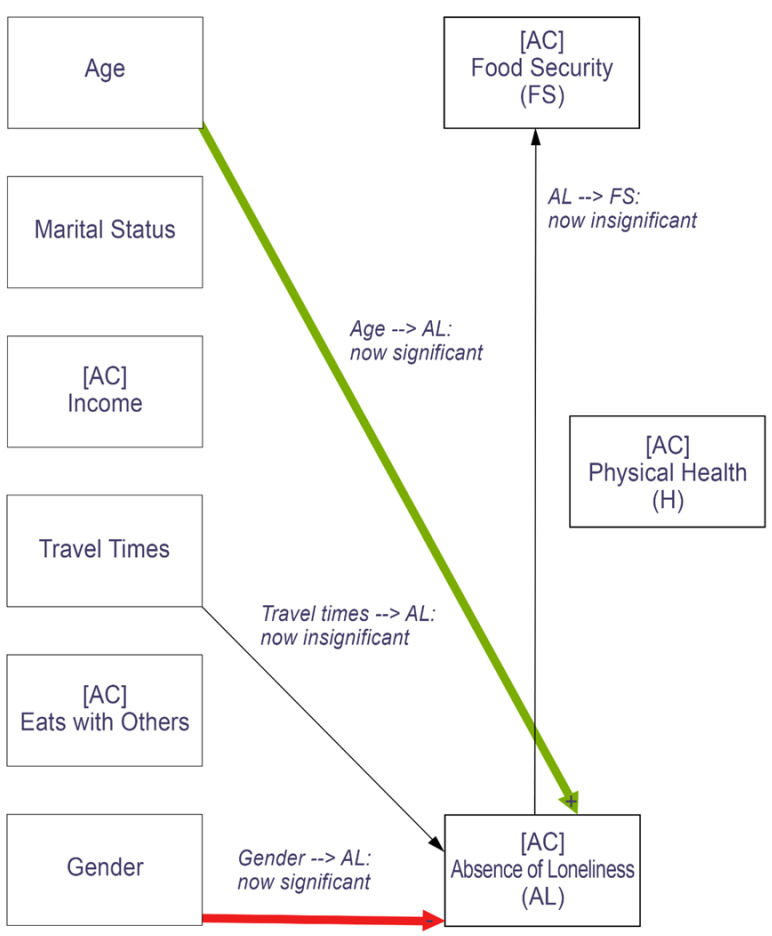
Path analysis results showing the differences in relationships after the onset of COVID-19 compared to the before-COVID path analysis. Thick lines represent significance at α = 0.05. Thin lines represent insignificance. Green represents positive relationships and red negative relationships.

**Figure 3 nutrients-14-05076-f003:**
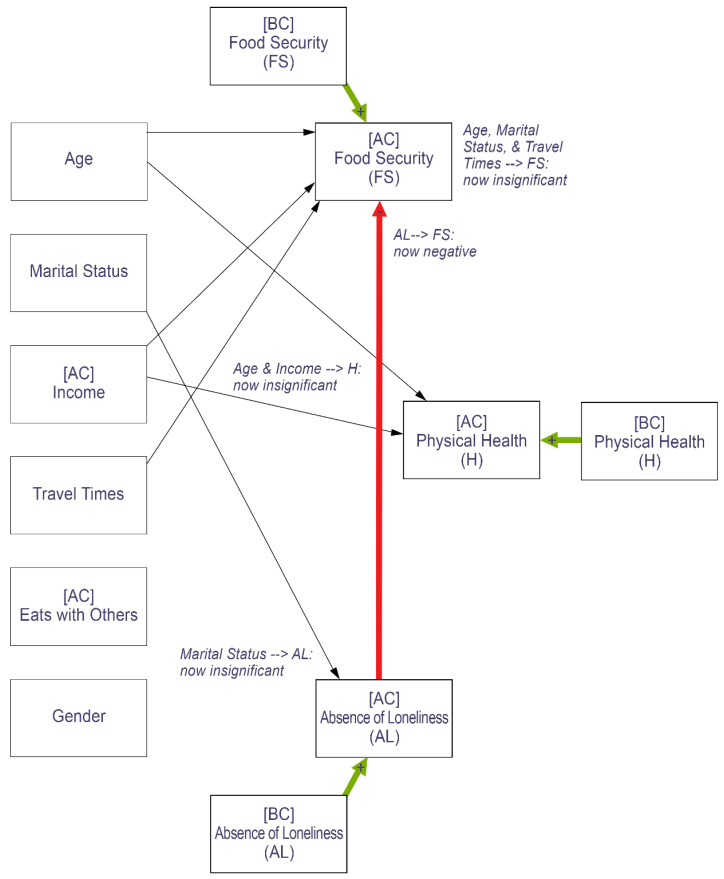
Path analysis results showing the differences in relationships after controlling for before-COVID food security, health, and loneliness (compared to the during-COVID path analysis). Thick lines represent significance at α = 0.05. Thin lines represent insignificance. Green represents positive relationships and red negative relationships.

**Table 1 nutrients-14-05076-t001:** Demographics of study counties.

County	% Adults Age 60+	Median Household Income	Poverty Rate	Food Insecurity Rate 2019	Food Insecurity Rate 2020	Below 130% Poverty
Crawford	20.10%	$41,662	16.50%	16.1%	15.4%	61%
Greene	19.80%	$51,613	13.50%	14.2%	13.3%	53%
Lawrence	20.60%	$53,610	10.50%	13.6%	12.3%	49%
Orange	19.80%	$47,917	14.10%	15.4%	16.3%	51%

**Table 2 nutrients-14-05076-t002:** Path analytical model variables and definitions.

Variable	Definition
Age	Numerical variable defining age of respondent in years.
Gender	Categorical variable defining respondent gender (0 = male; 1 = female). No respondents identified as anything other than male or female.
Marital Status	Categorical variable defining marital status (1 = married or living with a partner; 0 = widowed, divorced, separated, or never married).
Income	Ordinal variable defining respondent’s household monthly income class before COVID in USD. (0 = less than $1000; 1 = $1001–$1500; 2 = $1501–$2000; 3 = $2001–$2500; 4 = $2501–$3000; 5 = $3001–$3500; 6 = $3501–$4000; 7 = $4001–$4500; 8 = greater than $4500).
Travel Time	Numerical variable defining total one-way household travel time to supermarkets, grocery stores, or other stores accepting SNAP in the study region and counties bordering the study region.
Eats with Others	Ordinal variable defining the frequency of eating with others (4 = always; 3 = sometimes; 2 = half the time; 1 = seldom; 0 = never).
Food Security (FS)	Numerical variable reflecting the inverse of the total score of the respondent on the U.S. Six-Item Food Security Survey Module. The unadjusted scores range from 0 to 6, and the scale scores (used for statistical procedures based on the USDA guidelines) range from 0 to 8.48. For this analysis, 8.48 indicates the highest level of food security, and 0 indicates the lowest level of food security.
Physical Health (H)	Numerical value denoting the number of physically healthy days out of the past 30 days reported by the respondent. This is indexed by the inverse answer to the first question of the HRQOL-4 which asks, “Now thinking about your physical health, which includes physical illness and injury, for how many days during the past 30 days was your physical health not good?”. We subtracted the response from 30.
Absence of Loneliness (AL)	Ordinal variable reflecting the inverse of the sum of the Three-Item Loneliness Scale. This score ranges from 3–9, with 3 indicating the most loneliness and 9 indicating no loneliness.

**Table 3 nutrients-14-05076-t003:** Sample characteristics.

Variable	% or $\bar{x}$	Variable	% or $\bar{x}$
Gender			
Female	71.16%	Male	28.84%
White/Caucasian	96.5%		
Age	71.11		
Education			
Less than high school	2.10%	Some college	22.54%
Some high school	4.36%	College degree	14.65%
High school diploma/GED	38.32%	Post-college degree	11.12%
Trade certification	6.91%		
Marital Status			
Married or living with partner	65.31%	Divorced or separated	11.18%
Widowed	20.51%	Never married	2.99%

**Table 4 nutrients-14-05076-t004:** Changes after the onset of COVID-19.

Variable	Before COVID	During COVID	P
Monthly Income			0.000
≤$1000	7.13%	8.28%	
$1001–$1500	13.61%	15.25%	
$1501–$2000	12.64%	12.21%	
$2001–$2500	11.35%	11.15%	
$2501–$3000	9.89%	9.59%	
$3001–$3500	8.27%	8.52%	
$3501–$4000	7.78%	6.64%	
$4001–$4500	8.91%	8.69%	
>$4500	20.42%	19.67%	
SNAP Participation	2.63%	4.56%	0.000
Household Size	1.62	1.63	0.795
Eats with Others			0.000
Never	6.49%	40.72%	
Seldom	47.07%	35.86%	
Half the time	22.57%	5.35%	
Sometimes	13.60%	8.90%	
Always	10.28%	9.17%	
Travel Times (minutes, one-way)	16.65	17.03	0.361
Food Security Scale Score	8.19	8.15	0.019
Food Insecure	12.56%	13.28%	0.000
Physically Healthy Days (out of the past 30 days)	25.59	25.13	0.000
(Absence of) Loneliness Score	8.16	7.31	0.000

## Data Availability

The data presented in this study are openly available in Indiana University’s IUScholarWorks at http://hdl.handle.net/2022/26864.

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
