# Peer review of "The Interactions of Food Security, Health, and Loneliness among Rural Older Adults before and after the Onset of COVID-19"

_nutrients, 2022, doi:10.3390/nu14235076_

Round 1

Reviewer 1 Report

Thank you for the opportunity to review this manuscript, examining the interactions between the COVID-19 pandemic and health in older adults. This topic is of high importance as the study addresses an important literature gap in understanding the wider effects of the COVID-19 pandemic.

I have outlined a few comments below to help clarify several aspects of the manuscript:

Abstract/overall:

·        Overall an interesting study. Studies examining the effects of the pandemic are still uncommon so it is good to see a study with robust methods

Introduction:

·        Line 35, citation required for claim that rural communities most affected by food insecurity, poor health and loneliness associated with the pandemic

·        Line 41, citation required for closing of congregate meal sites

·        Line 62, add brief definition of “rural” as this term is used throughout

·        Line 74, add brief definition of “SNAP” for international readers

·        Paragraph 81, mention the relationship between low income and food insecurity, and the relationship between low income and poor health as all these factors are related

·        Line 103, citation for relationship between social interactions quality of life

·        Line 106, citation for relationship between loneliness and lack of desire to cook meals

·        Line 112, citation for rural residents experiencing more loneliness because homes are further apart

Methods:

·        Line 122, citation for previous studies examining quality of life

·        Line 182-183, add brief explanation of what the USDA Six-item Household Food Security Survey Module, the Healthy Days Core Model Questionnaire, and the UCLA Three-Item Loneliness Scale are

Discussion:

·        Line 341, change to past tense “found”

·        Line 371, is it just the association between marriage and eating together that allows better diet quality in older males? Older males usually have less confidence with cooking and less food preparation skills which also leads to poorer diet quality

·        Line 393, citation for relationship between loneliness and poor diet

·        Paragraph 383, add how older populations were the most vulnerable to COVID-19 infections and so were generally very compliant with stay at home recommendations

·        Paragraph 412, further limitation – this study was also conducted early in the pandemic and may not be representative of these health factors as the pandemic and government health recommendations evolved

Conclusion:

·        Line 430, citation for rural communities being the most vulnerable to negative impacts on quality of life

·        Line 442, suggest rephrase of this sentence. When speaking about older rural women earlier in the paper it is suggested that they have a social support network that they are removed from due to the pandemic, hence their loneliness. Then in the conclusion it appears that you are saying that older rural women are “very isolated” usually. Earlier in the paper it appeared that older men are the most lonely usually.

·        Line 443, suggest rephase. “Increased loneliness disconnected this population from food security pathways and created a stronger connection with physically unhealthy days and a weaker relationship with food security during the pandemic” – is it the loneliness that has disconnected them, or the pandemic itself forcing them to stay at home? The questions asked in your survey also did not address their engagement with “food security pathways” such as foodbanks

Author Response

Thank you for your helpful feedback and comments regarding our manuscript. “The interactions of food security, health, and loneliness among rural older adults before and after the onset of COVID-19”. Your time is truly valued and appreciated, and we have updated the manuscript accordingly. We have included all the recommended citations and definitions (rural, SNAP). We added a brief mention of the associations between low income and food insecurity and poorer health outcomes in paragraph 85 as suggested. We also added a brief explanation of the food security, healthy days, and loneliness scales to the methods section as recommended. We changed “find” to “found” in the discussion and added the recommended sentences regarding older adults and compliancy with stay-at-home recommendations. Regarding line 371 and marriage and older males—this sentence was not meant to only refer to males, but to older adults in general. In addition, it was referring to loneliness and food insecurity, not particularly diet quality. We have made this paragraph clearer and eliminated some of the wording that may have caused the confusion. We agree that a limitation to the study is the earliness of the survey after the onset of the pandemic. Thus, we have added this to the limitations section as recommended. In the conclusion, we rephrased the sentences mentioning women and the connections between the pandemic and the outcome variables. We feel they are now clearer and better represent the study findings.

Thank you, again,

Mecca Burris and team

Reviewer 2 Report

The manuscript presents the interrelationships among food security, physical health, and loneliness among rural older adults before the onset of and during the COVID-19 pandemic. The research hypotheses concerned that food security, physical health, and the absence of loneliness would positively associate with one another. Moreover, it was also hypothesized that age, marital status, income, travel times to stores, and eating with others would be associated with food security and physical health. Furthermore, the authors analyzed how these interactions and overall quality of life may have changed as a result of the COVID-19 pandemic. Generally, the manuscript is presented well-structured, and the experiments are designed appropriately to test the hypothesis.

However, the Reviewer found some concerns regarding the meager participation rate in surveys (29.6%). It has to be commented on as limitations in relation to the possibility of extending the conclusions to the population.

Minor: line 404 double dot

Author Response

Thank you so much for your valuable feedback and support for our manuscript titled "The Interactions of Food Security, Health, and Loneliness among Rural Older Adults Before and After the Onset of COVID-19". As recommended, we have added a line to the limitations section of the paper that speaks to the response rate and caution in regards to the representation of the findings.

Thank you, again, for your time and comments.